# An Overview of Selected Bacterial Infections in Cancer, Their Virulence Factors, and Some Aspects of Infection Management

**DOI:** 10.3390/biology12070963

**Published:** 2023-07-05

**Authors:** Amitabha Ray, Thomas F. Moore, Rajashree Pandit, Adam D. Burke, Daniel M. Borsch

**Affiliations:** 1College of Medical Science, Alderson Broaddus University, 101 College Hill Drive, Philippi, WV 26416, USA; mooretf@ab.edu; 2D’Youville University, Buffalo, NY 14201, USA; panditr@dyc.edu; 3Excela Health System, Latrobe, PA 15650, USA; aburke2@excelahealth.org; 4Lake Erie College of Osteopathic Medicine at Seton Hill, Greensburg, PA 15601, USA; dborsch@lecom.edu

**Keywords:** cancer-related bacteria, neoplastic pathology, virulence factor, secondary infection, antimicrobial stewardship, cancer mitigation

## Abstract

**Simple Summary:**

Bacterial involvement in cancer can be grouped into three categories: (i) direct association of bacteria in the processes of cancer development, (ii) secondary infection as a consequence of a patient’s weakened immune system, and (iii) utilization of bacteria in cancer therapeutics. Regarding a bacterial etiological or cancer-causing role, effective prevention strategies could be formulated after obtaining a precise understanding of the pathological mechanisms. Therefore, looking into the virulence factors of relevant bacterial species is critical in support of altering treatment plans with a proactive approach. This review has attempted to analyze both clinical/epidemiological and laboratory studies to understand bacterial mechanisms in cancer formation, particularly the virulence factors of *Helicobacter pylori* and *Chlamydia*. From secondary infections in cancer patients, microorganisms such as *Pseudomonas aeruginosa*, *Escherichia coli*, and *Klebsiella* species are commonly isolated. In general, these are highly diverse bacterial species and they have an exceptional ability to adapt and develop resistance against antimicrobial agents. Using an improved understanding of the pathogenic mechanisms of bacteria, it is necessary to review the topic of antimicrobial stewardship to optimize patient outcomes. Analysis of prophylactic antibiotic usage must also be addressed. Recent findings certainly support further examination of bacterial products that can be evaluated for their utilization as anti-cancer therapeutic agents.

**Abstract:**

In cancer development and its clinical course, bacteria can be involved in etiology and secondary infection. Regarding etiology, various epidemiological studies have revealed that *Helicobacter pylori* can directly impact gastric carcinogenesis. The *Helicobacter pylori*-associated virulence factor cytotoxin-associated gene A perhaps plays an important role through different mechanisms such as aberrant DNA methylation, activation of nuclear factor kappa B, and modulation of the Wnt/β-catenin signaling pathway. Many other bacteria, including Salmonella and Pseudomonas, can also affect Wnt/β-catenin signaling. Although *Helicobacter pylori* is involved in both gastric adenocarcinoma and mucosa-associated lymphoid tissue lymphoma, its role in the latter disease is more complicated. Among other bacterial species, *Chlamydia* is linked with a diverse range of diseases including cancers of different sites. The cellular organizations of *Chlamydia* are highly complex. Interestingly, *Escherichia coli* is believed to be associated with colon cancer development. Microorganisms such as *Escherichia coli* and *Pseudomonas aeruginosa* are frequently isolated from secondary infections in cancer patients. In these patients, the common sites of infection are the respiratory, gastrointestinal, and urinary tracts. There is an alarming rise in infections with multidrug-resistant bacteria and the scarcity of suitable antimicrobial agents adversely influences prognosis. Therefore, effective implementation of antimicrobial stewardship strategies is important in cancer patients.

## 1. Introduction

Pathogenic bacteria possess a large number of molecules and diverse mechanisms, which act as virulence factors and avert the host defense systems in order to initiate the disease processes. Apart from cell surface structures and metabolic arrangements, bacteria secrete several substances such as enzymes, toxins, and exopolysaccharides—many of these secreted biomolecules can enter the host cells and are able to control or damage the intracellular signaling/structures [1]. All of these bacterial biomolecules ultimately contribute to virulence and may lead to treatment complexity in the clinical setting.

One of the most common infectious agents and also one of the most studied organisms is *Escherichia coli* (*E. coli*), a Gram-negative, non-spore-forming, flagellated bacillus belonging to the family *Enterobacteriaceae*. Pathogenic *E. coli* strains are subdivided into different categories according to their clinical manifestations such as enteropathogenic (EPEC), necrotoxigenic (NTEC), and uropathogenic *E. coli* (UPEC). The prevalence of urinary tract infections (UTIs) is common in society and hospital settings, and UPEC is the leading pathogen in both uncomplicated and complicated UTIs [2]. Besides cell surface virulence factors (like fimbriae and capsular lipopolysaccharide/LPS) that help adhesion to epithelium/urothelium, the invasion of host cells, and biofilm formation, UPEC releases a number of virulence factors, e.g., α-haemolysin (HlyA), cytotoxic necrotising factor 1 (CNF1), secreted autotransporter toxin (SAT), cytolethal distending toxin (CDT), and toll/interleukin-1 receptor domain-containing protein (Tcp) [3]. These virulence factors play a significant role in bacterial colonization and disease propagation.

An intriguing aspect of certain bacterial virulence factors is their link with carcinogenesis. It is thought that chronic inflammation induced by *E. coli* in inflammatory bowel disease could predispose colon cancer development [4]. Of note, from *Enterobacteriaceae* family—*E. coli* and *Klebsiella* spp., and another Gram-negative bacillus—*Pseudomonas aeruginosa* have been commonly isolated from cancer patients [5,6,7]. In addition, patients with cancer are more vulnerable to secondary bacterial infections, which are a major cause of morbidity and mortality among these patients. However, evidence also shows a connection between certain bacterial persistent infections and carcinogenesis, e.g., *Helicobacter pylori* in gastric cancer, *Salmonella typhi* in gallbladder cancer, and *Chlamydia pneumoniae* in lung cancer [8,9]. Bacterial products such as colibactin (from *Enterobacteriaceae*) and cytotoxin-associated gene A (cagA, from *H. pylori*) can induce neoplasia by a number of mechanisms such as modulation of Wnt signaling pathway or maintaining an inflammatory state [8]. Interestingly, on the one hand Salmonella spp. (of the *Enterobacteriaceae* family) are believed to promote colon cancer, but on the other hand these bacteria may be used as anti-cancer agents. The Salmonella genome encodes a wide range of virulence factors, which can alter various intracellular signaling pathways including the Wnt/β-catenin and transforming growth factor-β (TGF-β), thus contributing to tumorigenesis [10]. In contrast, it has been demonstrated in experimental models that different attenuated Salmonella strains (such as A1-R and Χ9241) can preferentially colonize solid tumors and prevent tumor growth [11]. Similarly, virulence factors of *P. aeruginosa* such as mannose-sensitive hemagglutinin (PA-MSHA), cyclodipeptides, and exotoxin are being evaluated to be used in anti-cancer management [12,13,14]. A comprehensive understanding of these virulence factors at the cellular and molecular levels is greatly helpful to approach various pathological issues as well as their potential therapeutic aspects.

## 2. *Helicobacter pylori* and Gastric Adenocarcinomas

*H. pylori* (formerly called *Campylobacter pylori*) is a Gram-negative, flagellated, motile, and spiral-shaped/helical bacillus that grows in the gastric mucosa layer of more than 50% of the human population [15]. This bacterium is microaerophilic, i.e., it is aerobic but requires lower environmental oxygen levels, and it generally creates an asymptomatic chronic state of inflammation, which could lead to gastritis, peptic ulcer, gastric adenocarcinoma, and mucosa-associated lymphoid tissue (MALT) lymphoma of the stomach. After the discovery of *H. pylori* in 1982 by Barry J. Marshall and John R. Warren [16], it took about 6 years to realize that this bacterium might cause neoplasia [17]. Subsequently, some investigators showed an increased risk of gastric cancer among *H. pylori* seropositive subjects [18,19].

In an interesting review, Chiba et al. described two major pathways for the development of gastric adenocarcinoma due to *H. pylori* chronic infection [20]. One of the most important virulence factors in the direct pathway is the cagA protein. Of note, *H. pylori* can be divided into two strains, cagA-positive and cagA-negative; the former has a strong association with gastric cancer development. Furthermore, *H. pylori*’s direct action on gastric epithelial cells can cause the activation of nuclear factor kappa B (NF-κB), induction of mutations of tumor suppressor p53, and aberrant DNA methylation.

A key pro-inflammatory NF-κB pathway has been shown to promote the neoplastic processes of gastric mucosa. A study on AGS, SNU-1, and HGC-27 gastric cancer cell lines revealed that the activation of NF-κB occurred in *H. pylori* infection, which again triggered the activation of another pro-inflammatory transcription factor—signal transducer and activator of transcription 3 (STAT3) [21]. Similarly, after analyzing 255 human gastric cancer specimens immunohistochemically, one study noticed a significantly positive correlation between NF-κB and STAT3 activation [22]. Another study that examined gastric antral biopsies from 35 *H. pylori*-infected gastritis cases with 15 *H. pylori*-negative controls observed a distinctly enhanced NF-κB expression in the *H. pylori*-infected cases [23]. A number of in vivo and in vitro studies also recorded that, in the stomach, NF-κB activation was induced by *H. pylori* infection [24,25,26]. Furthermore, many studies documented that *H. pylori* disrupted the p53 tumor suppressor pathway, and p53 alterations can be observed in lesions ranging from gastritis to gastric cancer [27]. In a study on patients with chronic gastritis, it was detected that *H. pylori*-related lesions expressed the mutant-type p53 [28]. In addition, a lower expression of p53 was noticed in gastric epithelium of *H. pylori*-infected cases [29]. In the same way, a study on gastric cancer patients found that 80% of *H. pylori*-positive cases had p53 mutation [30]. Studies on different gastric cancer cell lines, e.g., STKM2, AGS, SNU1, and HFE145 cells, also showed that *H. pylori* infection resulted in p53 inhibition [31]. On the other hand, epigenetic modifications such as aberrant DNA methylation in the promoter regions of genes result in the alteration of different cancer-related genes and inactivation of tumor suppressor genes. In gastric cancer, aberrant methylation occurs in genes associated with DNA repair (such as hMLH1 and MGMT), transcriptional regulation (such as HLTF), cell growth/differentiation (such as HoxD10 and NDRG2), cell cycle (such as p16), and apoptosis (such as BNIP3) [32]. After analyzing gastric mucosa in *H. pylori*-linked chronic gastritis, gastric cancer, and cases with pre-neoplastic lesions followed up for 10 years, Compare et al. concluded that global DNA hypomethylation is an initial feature in *H. pylori*-linked gastric tumorigenesis [33]. Interestingly, Zhang et al. revealed hypermethylation of the tumor suppressor MGMT in *H. pylori*-induced gastric cancer development [34]. Their study showed that cagA enhanced the hypermethylation of tumor suppressor genes by stimulating DNA-methyltransferase 1 (involved in DNA methylation) via activation of AKT―NF-κB.

It is known that *H. pylori*-associated gastric cancer has a strong connection with gastritis, especially chronic atrophic gastritis of the corpus or type AB gastritis, where Th1-type CD4 cells and their product interferon-gamma (IFN-γ) may play a critical role, along with other pro-inflammatory cytokines such as interleukin-1β (IL-1β), IL-6, and tumor necrosis factor-α (TNFα). Apart from cagA, the bacterium also has a number of virulence factors, including vacuolating cytotoxin A (vacA), blood group antigen-binding adhesin (babA), and sialic acid-binding adhesin (sabA) [35,36]. These virulence factors trigger different cell-proliferation-related signaling pathways, e.g., phosphatidylinositol 3-kinases (PI3K)–Akt, Janus kinase (JAK)–STAT, and extracellular signal-regulated kinases (ERK), which could promote carcinogenesis in uncontrolled circumstances. It is notable that virulence factors such as babA and sabA bind to different blood group antigens (ABO and Lewis) on the gastric epithelial surface. This correlates to Wang and colleagues’ observation that persons with blood group A were more susceptible to *H. pylori* infection [37]. Overall, several investigators have recorded a higher risk of gastric cancer in blood group A-positive individuals [38].

## 3. Gastric MALT Lymphomas

Among *H. pylori*-infected people, approximately 2–3% and 0.1% of individuals develop gastric adenocarcinoma and gastric MALT lymphoma, respectively [35]. MALT lymphomas in the stomach are extranodal B-cell marginal-zone lymphomas with an indolent disease course [39]. When considered separately, probably none of the virulence factors that are commonly associated with gastritis, peptic ulcers, and gastric carcinoma has any significant role in gastric MALT lymphomas [40,41]. However, the gene cluster encompassing iceA1 allele, sabA, and hopZ, along with ORF JHP950, has been found to be associated with the risk of gastric MALT lymphomas. Nevertheless, gastric MALT lymphoma-related *H. pylori* strains are perhaps less virulent than those involved in gastric carcinoma [41]. On the other hand, it has been hypothesized that chronic inflammatory/antigenic stimulation by persistent *H. pylori* infection could result in continued immune cell activation that might favor malignant transformation [42,43]. Fortunately, *H. pylori* eradication therapy at the initial stage and for low-grade lymphoma is associated with satisfactory outcomes (Table 1) [44,45,46,47].

Overall, the gastrointestinal tract is a common extranodal site for the development of lymphomas, although the incidence rate of primary lymphomas in the gastrointestinal tract is low and the majority of these lymphomas originate in the stomach (Figure 1). It has been already stated that gastric MALT lymphoma is a slow-growing non-Hodgkin lymphoma, which is commonly associated with *H. pylori* infection. Without appropriate therapeutic management, *H. pylori* infection causes chronic inflammation that leads to the proliferation of B-cells and T-cells in the gastric mucosa. Finally, such a long-established inflammation stimulates the formation of aberrant mucosa-associated lymphoid tissue that can turn into malignancy [48]. In general, the eradication of *H. pylori* rectifies this lymphoid tissue problem. On the other hand, genetic aberrations such as chromosomal translocations in gastric MALT lymphomas can cause resistance to *H. pylori* eradication treatment. In this context, the most common translocation is t(11;18)(q21;q21) and the fusion product is a powerful NF-κB activator [49]. In addition, t(1;14)(p22;q32) is associated with the advanced stage. It may be worth mentioning that proton pump inhibitors are primarily metabolized through cytochrome P450 (CYP) 2C19, so polymorphisms in CYP 2C19 can affect the treatment [50]. Furthermore, bacterial factors for treatment resistance include mutations in the V domain of the 23S rRNA (most commonly A2143G) that reduce the affinity for clarithromycin, and mutations in the PBP1A in cases with amoxicillin resistance. Patients who are refractory to *H. pylori* eradication therapy should undergo alternative regimens such as chemotherapy and radiotherapy [47]. Regarding the prognostic markers, studies have revealed different predictors for poor patient outcomes such as older age, advanced staging, elevated lactate dehydrogenase levels, and high-grade histological subtype [51,52].

Histological features of gastric MALT lymphomas include lymphoepithelial lesions and the presence of centrocyte-like cells, monocytoid cells, immunoblast-like cells, and plasma cells [53,54]. Moreover, reactive germinal centers (which might be infiltrated by lymphoma cells) with mantle zone obliteration could be observed. The immunophenotype is not specific—the neoplastic cells express CD19, CD20, CD22, and CD79a (i.e., pan B-cell markers) [55]. Interestingly, MALT lymphomas originate from mature post-germinal center B-cells, which are similar to the plasma cells [56]. Therefore, a number of patients displayed excess biosynthesis of monoclonal immunoglobulins (IgG or IgM), which may mimic conditions such as Waldenström macroglobulinemia [56,57]. On the other hand, gastric MALT lymphomas can increase the risk for the development of intestinal metaplasia (precancerous lesions) and subsequent gastric adenocarcinomas [58,59].

## 4. Colon Cancer—Connection with Bacterial Pathology

Gastrointestinal tract cancers represent more than one-fourth of the overall cancer incidence and the two major sites are the stomach and colon, which comprise nearly 60% of gastrointestinal tract cancers [60] (Figure 1). According to the World Health Organization, the most commonly diagnosed gastrointestinal malignancy in 2020 was colon cancer (almost 2 million cases). The risk factors for colon cancers are primarily associated with lifestyle and dietary habits such as obesity and the intake of processed/red meat and alcohol, apart from hereditary and chronic inflammatory conditions. However, a growing body of evidence suggests an important role of gut microorganisms and their imbalance (dysbiosis) in the development of colon cancers. In this connection, many studies have recorded the pathogenic effects of specific bacterial species such as *E. coli*, *Fusobacterium nucleatum*, *Bacteroides fragilis*, and *Streptococcus gallolyticus* (*Streptococcus bovis*) (Table 2) [61,62,63,64,65,66,67,68,69,70,71,72,73,74,75,76,77,78,79,80,81,82,83]. By appropriate assessment of these bacteria and their interactions in the pathological processes, suitable prevention strategies could be designed.

## 5. *Chlamydia* Species: Their Virulence Factors and Involvement in Different Diseases Including Cancer

The life cycle of *Chlamydia* is unique. *Chlamydiae* are non-motile, Gram-negative, obligate intracellular bacteria, and exist in two morphologically distinct forms: elementary body (EB) and reticulate body (RB). The chlamydial cell wall consists of an outer membrane (containing a single major outer membrane protein/MOMP, and LPS, but no detectable peptidoglycan) and an inner cytoplasmic membrane [84]. The EB is the metabolically inert, infectious, environmentally resistant extracellular form, which has some similarities with a spore. It is responsible for spreading the infection within the host and transmission to other susceptible persons/hosts. The EB is capable of binding and invading primarily the mucosal epithelium, particularly columnar cells. For instance, in the case of *Chlamydia trachomatis* infection, the target areas are usually the single-layer columnar cells or their transformation zone with the non-keratinizing squamous epithelium, which is typically found in the genital tract [85]. The EB can bind with various cell surface receptors/molecules, e.g., cystic fibrosis transmembrane conductance regulator (CFTR), β1 integrin, epidermal growth factor receptor (EGFR), 3′sulfogalactolipid, ephrin receptor A2, apolipoprotein E4, and platelet-derived growth factor receptor (PDGFR) [86]. On the other hand, RB is the non-infectious replicative form. The surface membrane of the target cell forms a vacuole around the EB after it enters that transforms into a metabolically active RB. As a result, the characteristic inclusion bodies, which contain replicating organisms/RB, are formed near the nucleus. Finally, the condensation of RB (or intermediate body) results in the formation of infectious EB and the death of the host cell.

Three pathogenic species for humans are *Chlamydia trachomatis*, *Chlamydia pneumoniae*, and *Chlamydia psittaci* (zoonotic transmission). The latter two species can cause pneumonia. However, *C. trachomatis* is divided into 3 variant strains (biovars), which are again subdivided into several serotypes (serovars). Serovars A–C are responsible for trachoma (trachoma biovar), whereas the sexually transmitted serovars D–K (genital tract biovar) can cause genital tract infections and serious complications such as pelvic inflammatory disease, infertility, and ectopic pregnancy [87]. In addition, *C. trachomatis* promotes human immunodeficiency virus infection and cervical cancer pathogenesis. On the other hand, the serovars L1–L3 (LGV biovar) causes lymphogranuloma venereum [87]. According to the Centers for Disease Control and Prevention (CDC), *Chlamydia* infections have chronically been the most frequent notifiable sexually transmitted disease in the United States, with over 1.5 million cases reported in 2020 alone. More than 60% of all reported cases were among persons aged 15–24 and with most hosts being asymptomatic. Thus, understanding the long-term effects of this bacterium in terms of virulence, and the pathological consequences such as cancer development, is imperative.

A vital step in chlamydial pathogenesis involves the mechanisms by which *Chlamydiae* acquire the necessary nutrients. The bacteria are unable to produce the essential components of energy transduction and nucleic acid biosynthesis, as well as a number of amino acid biosynthesis pathways. So, they acquire these vital nutrients (including ATP) by selectively redirecting transport vesicles and capturing intracellular organelles [88]. Regarding bacterial virulence properties, several factors/components may play an important role in disease severity [89]. These factors include MOMP, polymorphic outer membrane proteins (Pmp), type III secretion systems (TTSS), putative chlamydial cytotoxin, stress-response proteins, LPS and other glycolipids, plasmid (cryptic) gene product, macrophage infectivity potentiator protein, chlamydial adhesins/invasins, and metabolic processes such as iron sequestration and modulation of tryptophan availability. Obviously, in order to understand the pathological processes of *Chlamydia* precisely, it is necessary to identify the relevant chlamydial virulence factors and their specific roles in disease severity.

One of the most studied chlamydial virulence factors is plasmid glycoprotein 3 (Pgp3). Generally, a number of chlamydial species and strains carry a 7.5 kb plasmid that encodes 8 Pgps. However, it is assumed that the abovementioned plasmid protein is a contributing factor at least for the pathogenesis of *C. trachomatis* species [90]. Moreover, Sturdevant et al. have commented that the plasmid and inclusion membrane protein CT135 (chromosomal gene product) are important virulence factors [91]. Their experiments with plasmid-deficient and CT135-null *C. trachomatis* serovar D strains in C3H/HeJ mice showed a reduced infectious capability compared to wild-type bacteria. Interestingly, Borges et al. revealed that CT135 impacted the expression of many proteins which are supposed to play a key role in bacterial virulence, including CT456/Tarp [92]. Of note, CT456 is an effector protein delivered to the host cell by a TTSS, which is a feature of many Gram-negative pathogens, to alter cytoskeletal processes [93]. *Chlamydia* spp. can manipulate the host cytoskeleton component actin to facilitate their invasion, replication, and disease spread. By restructuring the host actin cytoskeleton, the chlamydial type III-secreted ‘translocated actin recruiting phosphoprotein’ (Tarp) effector possibly supports bacterial entry into host cells [94,95].

For adhesion to host cells, the infectious chlamydial EB requires adhesins that include various polymorphic membrane proteins/Pmp. It has been suggested that *C. pneumoniae* utilizes Pmp6, Pmp20, and Pmp21 [96]. On the other hand, Favaroni et al. found Pmp22D, Pmp8G, and outer membrane complex protein B (OmcB) as necessary adhesins during *C. psittaci* infection [97]. In a study on human endothelial cells, Niessner et al. observed that *C. pneumoniae* Pmp 20 and Pmp 21 increased pro-inflammatory IL-6 and monocyte chemoattractant protein-1 via NF-κB pathway [98]. Interestingly, *C. pneumoniae* Pmp21 can bind to EGFR and induce EGFR activation [96]. In addition, *C. pneumoniae* infection has been shown to be associated with activation of different signaling molecules such as PI3K, mitogen-activated protein kinase, or ERK (Table 3).

Iron is an essential nutrient for different pathogens as well as for the innate immune response [99]. However, chlamydial iron acquisition mechanisms are not clearly understood and may be affected by the host’s fluctuating iron status. For example, in female genital tract infections with *C. trachomatis*, variations in the levels of lactoferrin due to estrogenic alterations may change chlamydial iron availability [100]. On the other hand, iron chelator 2,2-bipyridyl can cause iron starvation in *C. trachomatis* [101]. Nevertheless, Pokorzynski et al. have hypothesized that iron is transported to chlamydial cells by the YtgABCD ABC-type metal permease complex, and YtgR is an iron-dependent transcriptional repressor, regulated by tryptophan availability [100,102,103].

*Chlamydiae* have a highly complex cell structure and hence they were originally considered to be viruses. Of note, poxviruses replicate in the host cytoplasm like non-viral intracellular pathogens such as *Chlamydia* or *Rickettsia*. Unlike *Rickettsia* spp., that are broadly divided only into a few categories such as the spotted fever and typhus groups, *Chlamydia* spp. can cause a diverse array of diseases. For example, *C. trachomatis* can cause eye diseases such as trachoma, inclusion conjunctivitis, and trichiasis [104]. It has been suggested that *C. pneumoniae* may contribute to a number of vascular disorders such as endothelial damage, angiogenesis, and atherosclerosis [105,106]. Interestingly, both *C. pneumoniae* and *C. trachomatis* have been shown to be involved in otitis media [107,108]. Furthermore, studies have documented that *Chlamydia* infection can increase the risk of cancer for several sites, e.g., lung, ovary, uterine cervix, vulva, and ocular adnexa lymphoma (Table 4) [109,110,111,112,113,114,115,116,117,118,119,120,121,122,123]. Perhaps, routine *Chlamydia* infection screening in susceptible populations is useful to prevent various complications.

## 6. *Pseudomonas aeruginosa* and Cancer

Studies from different geographical locations have documented that *P. aeruginosa* infections frequently occur among cancer patients [124,125,126]. In addition, the evidence shows a substantially higher mortality rate in cancer patients with *P. aeruginosa* infections [126,127]. Although it is generally believed that cancer patients with neutropenia are more susceptible to pseudomonas infections [128,129], after reviewing a considerable number of reports on cancer patients, Maschmeyer and Braveny did not find any distinct differences between neutropenic and non-neutropenic cases, or between cases with solid tumors and hematologic malignancies, in connection with the involvement of *P. aeruginosa* infections [126]. Apart from oncological cases, this opportunistic bacterium is commonly detected in other clinical situations such as immunodeficiency, burn, conditions linked with the use of medical instruments/devices, and cystic fibrosis. It may be worth mentioning that in cystic fibrosis approximately 60–80% of adults finally develop chronic *P. aeruginosa* infection, and usually this pulmonary infection persists indefinitely [130]. In addition, various poor clinical features such as worsening symptoms (such as cough), nutritional status, lung function, radiographic scores, and poor prognosis (such as end-stage lung disease) are more commonly noticed in patients with chronic *P. aeruginosa* infection than in those without.

Regarding *P. aeruginosa* bacteria-related conditions, it is known that several pathological states under primary immunodeficiency, such as common variable immunodeficiency and ataxia-telangiectasia, and secondary immunodeficiency, such as acquired immunodeficiency syndrome, are associated with an increased risk of cancer [131,132,133]. In general, the incidence of lymphomas is relatively higher in these patients. On the other hand, many studies have observed an increased risk of cancers in the gastrointestinal tract and other sites such as the thyroid and kidney among patients with cystic fibrosis [134,135,136,137]. Of note, cystic fibrosis is an autosomal recessive disease, which is caused by mutations in the CFTR gene, located on chromosome 7. The CFTR protein may act as a tumor suppressor, and its deficiency perhaps causes disruption of several cellular processes that could be linked with neoplastic processes, e.g., influence on immune reactions, inflammatory responses, and Wnt/β-catenin signaling [138]. Interestingly, *P. aeruginosa* can also affect the Wnt/β-catenin signaling pathway [139].

*P. aeruginosa* is a Gram-negative opportunistic bacillus which can survive in diverse environments and is considered an important pathogen for nosocomial infections. Bacterial virulence factors include several components, e.g., motility (which is associated with surface appendages such as flagella and pili), LPS (a component of the outer membrane), secretion systems (which render toxins and enzymes into the host’s intracellular or extracellular space), secondary metabolites (such as phenazines like pyocyanin—responsible for greenish color), quorum sensing (i.e., cell-to-cell communication system), and biofilm formation [140,141]. Pathological phenomena such as biofilm formation and chronic infection are associated with bacterial capabilities of adherence and motility. Furthermore, LPS can induce immune responses and CFTR. On the other hand, there are five secretion pathways: the type 1 secretion system (T1SS), T2SS, T3SS, T5SS, and T6SS. Exotoxin A belongs to the T2SS and is responsible for host cell death, while the T3SS is linked with the pathogenicity/severity of infections.

The management of *P. aeruginosa* infections has become a serious problem because of the bacteria’s very high capability to develop resistance against several antibiotics. Furthermore, indiscriminate use of antibiotics favors the emergence of multidrug-resistant *P. aeruginosa* strains. In general, the principal mechanisms by which *P. aeruginosa* can resist antibiotic molecules may be categorized into three groups, i.e., intrinsic, acquired, and adaptive resistance [142]. Examples of intrinsic resistance are reducing the permeability of the outer membrane, the development of efflux pumps that eject antibiotic molecules out of the bacterial cell, and the biosynthesis of inactivating enzymes for antibiotics. The acquired resistance can be attained by suitable mutations and/or horizontal transfer of antibiotic resistance genes such as plasmid-mediated conjugation and natural transformation, i.e., obtaining foreign genetic material. On the other hand, adaptive resistance includes biofilm formation in the involved tissue. The biofilm (i.e., matrix of extracellular polymeric substances comprising embedded bacteria, extracellular DNA, proteins, and polysaccharides) prevents antibiotic diffusion. By means of the protection of biofilm, *P. aeruginosa* can survive and cause recurrent infections.

Interestingly, accumulating evidence indicates a potential role for *P. aeruginosa* in cancer therapy. Since the initial observation of a bacterial anti-neoplastic role by German physician W. Busch in 1868 [143], the exploration of bacterial products/relevant mechanisms against cancer has been reported by many investigating groups throughout the world (Table 5) [144,145,146,147,148,149,150,151,152,153,154,155,156]. Different studies have identified a number of *P. aeruginosa*-released anti-cancer substances such as exotoxin A, mono-rhamnolipids, and azurin [157,158,159]. Overall, *P. aeruginosa*-linked anti-tumor mechanisms include the growth of bacteria in hypoxic regions of tumors, the production of toxins, modulation of host immune responses, and delivery of therapeutic genes that encode cytotoxic peptides or prodrug-converting enzymes [160] (Figure 2). In addition, it may be worth mentioning that *P. aeruginosa* toxins are favorable biomolecules for the construction of recombinant immunotoxins or chimeric proteins (such as monoclonal antibody plus bacterial toxin) for anti-cancer therapeutic strategy. Nevertheless, a greater understanding of the interactions between cancer cells and bacteria will definitely be useful for the development of novel anti-cancer management.

## 7. Systemic Cancer Therapy—Immunity and Infection

It is believed that after the transformation of a normal cell to malignancy, there is a need for the initial cancer cell(s) to evade the barriers/attacks from the host’s immunosurveillance mechanisms in order to survive and grow. In this process, a dynamic tumor microenvironment is created wherein active interactions occur between different biological constituents, e.g., cancer cells, immune cells, various stromal cells, extracellular matrix, blood vessels, and even the nervous system [161]. However, according to the circumstances, cancer cells may switch to a quiescent phase or state of dormancy [162]. On the other hand, cancers can progress to the advanced state, metastasize to different body systems, and may cause cachexia. Cancer cells spreading into the bone marrow clearly weakens the immune system. Similarly, various cancer chemotherapeutic agents can cause bone marrow suppression, neutropenia, immunosuppression, and an increased risk of infection [163]. In cancer-associated cachexia, the pathological processes affect different cytokines (e.g., TNFα, IFN-γ, and IL-6) including adipokines and myokines, their signaling pathways, gut microbiota and gut hormones, immune cells, immune checkpoints, the immune–metabolic axis, and the nervous system [164]. In a study on colon cancer patients, Zhang et al. found that cachectic patients had a significantly higher rate of bacterial DNA fragments in serum than non-cachectic patients and healthy controls. Furthermore, the former group had higher levels of IL-1α, IL-6, IL-8, and TNFα [165]. In an interesting review, Herremans et al. discussed the alterations of microbiota (dysbiosis) and their links with gut barrier dysfunction, pathways of systemic inflammation, and muscle wasting in cancer cachexia [166].

Regarding immunomodulatory therapy against cancer, vaccines are currently available for protection against human papillomavirus (HPV) and hepatitis B virus (HBV); HPV is primarily linked with cervical cancer and HBV infection can initiate liver cancer. Different reports have documented that these two vaccines are generally well tolerated among populations throughout the world [167,168,169,170,171]. In systemic therapy against cancer, there is some overlap between immunotherapy and targeted therapy. Clinically, monoclonal antibodies against the human epidermal growth factor receptor-2 (HER2), e.g., trastuzumab, are commonly used in cancers that overexpress HER2, particularly in breast cancer [172]. Of note, HER2 belongs to the EGFR family and frequently overexpresses in many cancers [173]. In a meta-analysis on 8669 breast cancer patients receiving trastuzumab, Jackson et al. noticed several adverse effects including increased susceptibility to infections (particularly respiratory tract infections) [174]. On the other hand, in treatment with immune checkpoint inhibitors, toxicities can affect any body systems, which includes inflammation of different organs such as pneumonitis, hepatitis, vasculitis, as well as neutropenia, and anemia [175]. In this context, reports have documented different opportunistic bacterial infections such as *P. aeruginosa* and *Clostridium difficile*, apart from fungal and viral infections [176,177].

## 8. Antimicrobial Stewardship in Cancer Patients

As mentioned before, patients with solid tumors or hematological malignancies frequently develop infections. Of note, roughly more than 90% of all cancers are solid tumors. Nevertheless, the increased risk of infections in cancer patients could be due to tumor-related factors or therapy-associated complications, which include neutropenia, damage to biological barriers (such as integument and mucosa), obstruction of physiological passages, patient’s age, nutritional status, as well as surgical procedures and instrumentations for diagnosis or therapy [178]. In general, neutropenia is common in hematological malignancies, but this problem can also happen as a consequence of chemotherapy, radiotherapy, and bone marrow metastasis. On the other hand, disruption of mucosal surfaces can occur due to medical devices such as catheters or treatment-linked mucositis (chemotherapy or radiotherapy), and the common sites of infection are the respiratory tract, gastrointestinal tract, and urinary tract. It is noteworthy that several risk factors for infection commonly exist in the same patient.

Although the use of antibiotics is intimately connected with cancer care, an alarming rise in infections with multidrug-resistant bacteria and the scarcity of suitable antimicrobial agents are important problems in the treatment of cancer patients. Among patients with cancer, isolated drug-resistant bacteria include *E. coli*, *Klebsiella* spp., *P. aeruginosa*, viridans group streptococci, and coagulase-negative staphylococci [179]. It may be worth mentioning that in immunocompromised patients with neoplastic diseases, the most common complications are secondary bacterial infections, particularly bloodstream infections, which are also responsible for significant morbidity, mortality, and economic burden [180]. Considering the emergence of multidrug-resistant organisms, there is a need to formulate better strategies in order to prevent the development and spread of multidrug-resistant bacteria, as well as to carefully use the currently available antimicrobial drugs (which is referred to as antimicrobial stewardship) [181].

In a recent report from Japan, the investigators had studied the effectiveness of antimicrobial stewardship on 32,202 patients during 2018–2021 at a cancer hospital [182]. They observed a declining trend of methicillin-resistant *Staphylococcus aureus* and multidrug-resistant *P. aeruginosa* infections. In addition, the integration of the antimicrobial stewardship program (ASP) and facility of infectious disease consultations (IDC) decreased the use of carbapenem (a β-lactam class antibiotics) without negative patient outcomes. Finally, they concluded that the application of their method might support appropriate cancer management. In another study that was conducted during 2009–2017 in a cancer department in Spain, the investigators noticed that the combination of IDC and ASP benefited antibiotic use in cancer patients, and this improvement was associated with a decrease in mortality due to bacterial infections [183]. Likewise, a prospective cohort study in Brazil during 2009–2011 analyzed 307 episodes of chemotherapy-induced febrile neutropenia in 169 patients [184]. Interestingly, the study revealed that adherence to the ASP was correlated with decreased mortality rates.

Among cancer patients receiving chemotherapy, febrile neutropenia is a serious complication and is negatively associated with overall prognosis, including mortality [185]. It is worth mentioning that the major threats to neutropenic patients are infections with multidrug-resistant Gram-negative bacteria, particularly with pathogens such as *Klebsiella* spp., *E. coli*, and *P. aeruginosa* [186]. In addition, about one-third of cases of febrile episodes in patients with neutropenia are associated with bacteremia (i.e., presence of bacteria in the circulation). Interestingly, in developed countries, the predominant pathogens are Gram-positive bacteria, whereas mortality is higher in Gram-negative bacteremia [187].

Kouranos and his colleagues reviewed lung cancer cases where the patients were treated with prophylactic antibiotics and they noticed that febrile neutropenia, infections, and the duration of hospitalization were significantly reduced [188]. The authors concluded that prophylactic antibiotics use appears to be efficacious and may be used as a prevention strategy for chemotherapy-induced neutropenia in patients with lung cancer. In general, quinolones (such as levofloxacin and ciprofloxacin) and trimethoprim-sulfamethoxazole are the commonly used prophylaxis, although their prophylactic use is debatable. In another study, 113 primary randomized or quasi-randomized trials on patients with cancer receiving chemotherapy or undergoing hematopoietic stem cell transplantation with expected neutropenia were reviewed [189]. The authors observed that the prophylactic use of a quinolone, trimethoprim-sulfamethoxazole, or a cephalosporin regimen decreased bacteremia. Furthermore, they noted that the use of quinolone/fluoroquinolone was not significantly related to higher infection rates of *C. difficile* or invasive fungal pathogens. Overall, different studies have documented that antibiotic prophylaxis can decrease the episodes of febrile neutropenia and infection-linked deaths in patients receiving chemotherapy [190]. Additionally, a study conducted by Itoh et al. found that an oral switch from intravenous antimicrobial therapy was beneficial in preventing catheter-related bloodstream infection with methicillin-sensitive *S. aureus* in cancer patients [191].

Like antibacterial management, the antifungal prophylaxis strategy has been employed in several places to control invasive fungal infections in high-risk hematologic cases such as acute myeloid leukemia and myelodysplastic syndrome [192,193,194]. Of note, *Candida*, *Aspergillus*, and *Cryptococcus* species are common fungal pathogens that cause invasive infections in immunocompromised conditions, including cancer [195]. Furthermore, after reviewing the data of a number of trials, Robenshtok et al. revealed that antifungal prophylaxis significantly reduced the mortality in patients after chemotherapy [196]. Nevertheless, in the prophylactic use of antibacterial and antifungal agents, reasonable clinical practice guidelines are required for appropriate decision making, optimization, and selection of adequately targeted patients to achieve the maximum benefit [189,197]. Therefore, antimicrobial stewardship is vital in antibacterial and antifungal prophylaxis among cancer patients.

Interestingly, Aitken and his colleagues mentioned a number of antimicrobial stewardship strategies that have been effectively implemented in cancer patients [198]. Overall, important measures such as the improvement of clinical outcomes, the reduction of antimicrobial drug-related adverse effects/consequences, and a decrease in antimicrobial overuse are the primary goals of an ASP [198,199]. Effective collaboration between oncologists and the institutional ASP can establish successful antimicrobial stewardship for cancer patients and support substantial progress in cancer management.

## 9. Conclusions

Many bacteria are linked with the pathological processes of cancer in diverse ways. Apart from *H. pylori* infection, which clearly displays a direct etiological connection to tumorigenesis, the neoplastic role of other bacteria cannot be established so decisively. Moreover, the intricate interactions between specific bacteria and the target cells of our body, as well as the exact nature of different virulence factors, are chiefly unidentified. Oppositely, a number of bacterial products are being evaluated for their utilization as anti-cancer therapeutic agents. Therefore, on the one hand certain virulence factors are pathognomonic for the disease course, but on the other hand these biomolecules could play a key role in treatment strategies. While research continues, clinicians should be ever diligent in having patients participate in screening testing, especially those who are at risk for high-virulence pathogens. By eliminating these pathogens early in a disease process, cancer management can be improved. Secondary bacterial infections among cancer patients have a major impact on the overall prognosis and patients’ survival due to the immunosuppressive nature of malignancy and related chemotherapy/radiotherapy. Moreover, along with bacterial infections, an alarming increase in antibiotic resistance is a great challenge for patients’ wellness. Currently, the careful use of antibiotics is the sole method to provide the relief of cancer patients.

## Figures and Tables

**Figure 1 biology-12-00963-f001:**
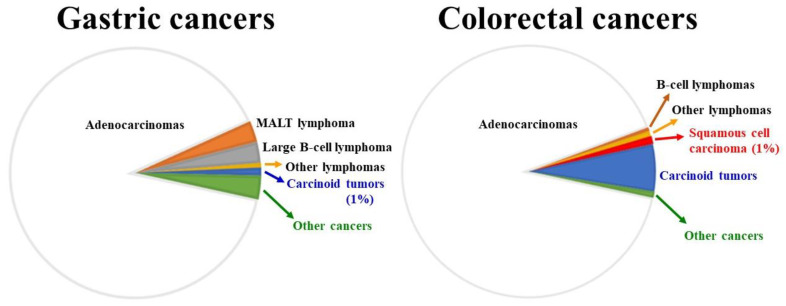
Comparison between adenocarcinomas and lymphomas for the two most common gastrointestinal sites for cancer development. Roughly 2 million cases of colorectal cancers and 1.1 million cases of gastric cancers were diagnosed in 2020 (source: World Health Organization). Both gastric and colorectal adenocarcinomas: ~90%; gastric lymphomas: ~6% (generally, MALT lymphomas are low-grade lesions, whereas large B-cell lymphomas are high-grade malignancy; both make up around 90% of all gastric lymphomas); colorectal lymphomas: ~1.2% (B-cell lymphomas: ~0.5%).

**Figure 2 biology-12-00963-f002:**
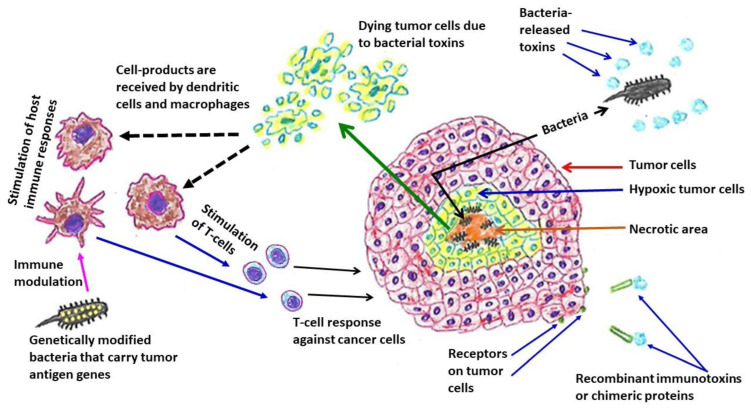
Bacteria-linked anti-tumor therapeutic strategies. Many bacteria, such as *Escherichia coli* and *Pseudomonas aeruginosa*, colonize in hypoxic regions of tumors. Immunotoxins or chimeric proteins are produced by combining bacterial toxins (such as *Pseudomonas aeruginosa* exotoxin A) with monoclonal antibodies or cytokines or growth factors, which can bind with the specific tumor cell surface molecules or receptors that are over-expressed. An example of genetically modified bacteria is *Pseudomonas aeruginosa*—mannose-sensitive hemagglutinin (PA-MSHA), a less virulent/attenuated strain.

**Table 1 biology-12-00963-t001:** Commonly followed *Helicobacter pylori* eradication therapy in gastric MALT lymphomas.

Treatment Course	Patients	Medication	Duration
First-line eradication therapy: triple therapy	General/initial cases	Proton pump inhibitor (40 mg twice a day) + clarithromycin (500 mg twice a day) + amoxicillin (1 g twice a day)	14 days
	For patients with hypersensitivity to amoxicillin	Proton pump inhibitor (40 mg twice a day) + clarithromycin (500 mg twice a day) + metronidazole (500 mg three times a day)	
Second-line eradication therapy: quadruple therapy	For patients with history of previous eradication therapy or unsuccessful triple therapy	Proton pump inhibitor (40 mg twice a day) + bismuth (300 mg four times a day) + tetracycline (500 mg four times a day) + metronidazole (500 mg three times a day)	14 days
Third-line eradication therapy: triple therapy (modified)	If the above regimens do not work	Proton pump inhibitor (40 mg twice a day) + amoxicillin (1 g twice a day) + levofloxacin (500 mg once a day)	10 days

Probiotics (such as *Lactobacillus* and *Bifidobacterium* species) could be used as an adjuvant therapy; Patients are evaluated by urea breath tests, 4–6 weeks after the completion of therapy.

**Table 2 biology-12-00963-t002:** Recent reports on commonly studied bacteria that may have a pathological role in the development of human colorectal cancer.

Investigators, Place of Study, and Study Design	Findings
Mirzarazi et al., 2022 [61], Iran Fecal samples from 20 patients and 50 control subjects	Increased expression of outer membrane protein A (OmpA) in the commensal *E. coli* B2 phylogenetic group from patients
Périchon et al., 2022 [62], France Fecal samples collected from patients with colorectal cancer at different stages (n = 81), adenoma (n = 23), and from normal colonoscopy (n = 25)	Significantly increased levels of *S. gallolyticus*, *F. nucleatum*, and *P. micra* in the colorectal cancer group. Increase of pks island at late-stage cancer.
Xie et al., 2022 [63], China Tumor and adjacent noncancerous tissues were collected from 184 colorectal cancer patients	Cancerous tissues had higher levels of *F. nucleatum* than adjacent noncancerous tissues. Disease-free survival was significantly longer in the *F. nucleatum*-negative, younger age, and TNM stage I–II groups.
Bertocchi et al., 2021 [64], Italy Cohort of 179 patients—10 years follow-up	*E. coli* disrupted the gut vascular barrier, which supported distant metastases.
Butt et al., 2021 [65], Europe Nested case–control study—pre-diagnostic serum samples from 442 cases and 442 controls	Sero-positivity to *E. coli* and ETBF (co-infection) was associated with colorectal cancer development.
Cuellar-Gómez et al., 2021 [66], Mexico Tumor, peri-tumoral, and normal tissue samples from 30 patients with colorectal cancer	*F. nucleatum* in cancer samples was significantly higher than in the normal mucosa samples (more than 10 cm from the cancer margins).
Khodaverdi et al., 2021 [67], Iran Colon tissue samples from 40 colorectal cancer patients and 40 healthy controls	*E. faecalis* and ETBF were significantly higher in colorectal cancer samples compared to control group.
Kong et al., 2021 [68], China Healthy controls (n = 30) and 30 pre-operative colorectal cancer serum/fecal samples, and also 18 fresh-frozen colonic tissues from controls along with 141 fresh-frozen tumor samples from patients were analyzed.	Among patients, a positive association between higher levels of fecal *F. nucleatum* and serum 12,13-EpOME; and increased levels of CYP2J2 in tumor samples also correlated with high *F. nucleatum* levels and worse overall survival in advanced disease.
Nardelli et al., 2021 [69], Italy Colon tissue/mucosa from 20 patients and 20 healthy controls	*E. coli*, *F. nucleatum*, *B. fragilis*, *S. intermedius*, and *G. haemolysans* were increased in patients’ tumor tissue and non-lesioned tissue.
Pignatelli et al., 2021 [70], Italy Samples from oral, cancer tissue, and adjacent non-neoplastic mucosa from 36 patients with colon cancer	There was an association between the *F. nucleatum* quantity in the oral and cancer tissue, and a significant association between the *F. nucleatum* abundance in cancer and staging.
Iyadorai et al., 2020 [71], Malaysia Mucosal colon tissues from 48 patients and 23 healthy controls	pks+ *E. coli* was more often isolated from tissues of patients.
Zamani et al., 2020 [72], Iran Mucosa biopsies from 68 patients with precancerous and cancerous lesions and 52 healthy controls	*B. fragilis* was more abundant in the patients’ samples than healthy controls. The bft gene was detected in 47% and 3.8% of the patients and controls, respectively.
Alomair et al., 2018 [73], Saudi Arabia Mucosal microbial samples were collected from 29 colorectal cancer patients along with matched controls, and DNA samples were used in the study	Fusobacterium and *B. fragilis* were more common in the carcinoma group.
Kwong et al., 2018 [74], Hong Kong Retrospective population-based cohort study during a 10-year period—13,096 adult patients with bacteremia	Increased risk of colorectal cancer in patients with bacteremia from intestinal microbes such as *B. fragilis*, *S. gallolyticus*, *F. nucleatum*, Peptostreptococcus spp, and *C. septicum.*
Proença et al., 2018 [75], Brazil DNA and/or RNA extracted from tumor and adjacent normal tissues from 27 colorectal adenoma and 43 colorectal cancer cases	Compared to matched normal tissue, excess *F. nucleatum* was detected in adenoma and more markedly in cancerous tissue. Moreover, KRAS mutations were more commonly seen in cancer samples infected with *F. nucleatum.*
Mehta et al., 2017 [76], USA Prospective cohort study—using data from the Nurses’ Health Study and the Health Professionals Follow-up Study; 1019 incident colorectal cancer cases over 26–32 years follow-up	Prudent diets rich in whole grains and dietary fiber were associated with a lower risk for *F. nucleatum*-positive cancer.
Xu and Jiang 2017 [77], China Mucosa biopsy from 52 cases with invasive adenocarcinomas, 47 cases with colorectal adenomas, and 61 non-tumor colon (normal group)	Fusobacterium and *B. fragilis* were more common in the colorectal cancer group.
Mima et al., 2016 [78], USA The Nurses’ Health Study and the Health Professionals Follow-up Study—1069 colorectal carcinoma cases	*F. nucleatum* DNA amount in cancer tissue was associated with shorter survival.
Magdy et al., 2015 [79], Egypt Prospectively enrolled 213 patients and 248 healthy controls	EPEC was significantly higher in patients. Patients’ *E. coli* differed serotypically and genotypically from *E. coli* in normal persons.
Bonnet et al., 2014 [80], France Tumors and mucosa of patients (n = 50) and the healthy mucosa of diverticulosis (controls, n = 33)	Compared to normal tissue, a higher level of mucosa-associated and internalized *E. coli* in tumors, and an association between poor prognostic factors (TNM stage) and *E. coli* colonization in mucosa were observed.
Kohoutova et al., 2014 [81], Czech Republic Mucosal biopsies from colorectal adenoma (n = 30), colorectal cancer (n = 30), and healthy controls (n = 20)	Higher production of colicins and microcins, and a higher frequency of *E. coli* phylogroup D, were found among patients with colorectal cancer.
Kostic et al., 2013 [82], USA Fecal samples from healthy subjects (n = 30), colonic adenomas (n = 29), and colorectal cancer (n = 27)	Fusobacterium spp. were enriched in stool samples from colonic adenoma and carcinoma patients compared to healthy subjects.
Rubinstein et al., 2013 [83], USA Analysis of fadA gene (*F. nucleatum* virulence factor) in colon specimens from normal individuals (n = 14), normal tissues from patients with precancerous adenomas (n = 16), precancerous adenomas (n = 16), normal tissues from patients with adenocarcinomas (n = 19), and adenocarcinomas (n = 19)	The fadA gene levels in the colon tissue from patients with adenomas and cancer were higher compared with normal subjects.

*E. coli* phylogroups: the species can be divided into 7 major phylogroups—A, B1, B2, C, D, E, and F. Gene cluster pks: *E. coli* B2 phylogenetic group has the polyketide synthetase (pks) genomic island that encodes for the biosynthesis of colibactin, which is responsible for DNA damage, chromosomal instability, and mutations. Microcins: low-molecular-weight peptidic toxins/bacteriocins with antimicrobial effects, which are released to inhibit other bacteria. *B. fragilis*: *Bacteroides fragilis*; BFT or bft gene: *Bacteroides fragilis* toxin; *C. septicum*: *Clostridium septicum*; Colicins: protein toxins/bacteriocins; CYP2J2: cytochrome P450 2J2; 12,13-EpOME: 12,13-epoxyoctadecenoic acid (synthesized through CYP2J2 activity); *E. coli*: *Escherichia coli*; *E. faecalis*: *Enterococcus faecalis*; EPEC: enteropathogenic *Escherichia coli*; ETBF: enterotoxigenic *Bacteroides fragilis*; *F. nucleatum*: *Fusobacterium nucleatum*; FadA adhesion gene: virulence factor of *F. nucleatum*; *G. haemolysans*: *Gemella haemolysans*; *P. micra*: *Parvimonas micra*; *S. gallolyticus*: *Streptococcus gallolyticus*; *S. intermedius*: *Streptococcus intermedius*; TNM: tumor–node–metastasis.

**Table 3 biology-12-00963-t003:** Selected cell-signaling pathways that are linked with bacterial infections and neoplastic processes.

Signaling Pathways or Relevant Biomolecules	Short Description
Akt or protein kinase B	A serine/threonine kinase, Akt plays an important role in cell cycle progression, cellular growth and survival, inhibition of apoptosis, and angiogenesis. The Akt signaling pathway is also associated with a number of biological molecules/processes, e.g., cytokine/growth factor receptors, integrins, PI3K, and immune mechanisms.
Cystic fibrosis transmembrane conductance regulator (CFTR)	A cyclic AMP-dependent chloride channel protein, CFTR may have a negative association with the NF-κB signaling pathway and thus cancer cell proliferation.
Epidermal growth factor receptor (EGFR)	A transmembrane glycoprotein receptor with tyrosine kinase activity, EGFR is involved in cell proliferation and survival. Over-expressions of EGFR and its family partner HER2 are noticed in several cancer types.
Extracellular signal-regulated kinase (ERK)	A member of MAP kinase family, ERK is activated through a sequential phosphorylation pathway, and associated with cell division and growth.
Janus kinase (JAK)	A non-receptor tyrosine kinase, JAK is linked with cytokine signaling (including IL-6 and IFN-γ), phosphorylation of the receptors, and activation of other signaling molecules such as STAT. In the development and progression of several cancers, JAK plays a significant role.
Mitogen-activated protein kinase (MAP kinase)	Three major groups of these serine–threonine kinases are ERK, c-Jun amino-terminal kinase (JNK), and p38 MAP kinase. They regulate various cellular activities such as cell cycle progression, cell proliferation, survival and death.
Nuclear factor kappa B (NF-κB)	A transcription factor, NF-κB participates in a wide range of pathophysiological phenomena, e.g., inflammation, cell proliferation and survival, immune responses, and a number of health disorders including cancer. Moreover, NF-κB is influenced by various stimuli such as IL-1β, IFN-γ, and TNFα.
Phosphatidylinositol 3-kinase (PI3K)	A cell-membrane-associated lipid kinase, PI3K can work with a number of signaling molecules such as Akt for cell growth and survival. The PI3K signaling is connected with several growth factor receptors such as EGFR and PDGFR, and pro-inflammatory cytokines such as IL-1β, IL-6, and TNFα.
Platelet-derived growth factor receptor (PDGFR)	A receptor tyrosine kinase, PDGFR is associated with the growth of mesenchymal cells. Its abnormal signaling is connected with a number of diseases, e.g., inflammation, atherosclerosis, pulmonary fibrosis, and cancer.
Signal transducer and activator of transcription (STAT)	A transcription factor that is connected with JAK and signal transduction by various growth factors/cytokines and hormones, e.g., IL-6, IFN-γ, and growth hormone, in order to control cellular processes such as cell proliferation, survival, and differentiation.
Transforming growth factor β (TGF-β)	A pleiotropic cytokine, TGF-β functions in the regulation of cell growth in both positive and negative manner. Different signaling pathways, e.g., PI3K, Akt, and MAP kinase, are associated with TGF-β―mediated functions.
Tumor suppressor p53	A transcription factor, p53 is activated by stress signals and its binding with DNA initiates the transcription of genes that are linked to various cancer-preventive cellular phenomena such as cell cycle arrest and cell death.
Wnt/β-catenin	This pathway plays an important role in cell proliferation, apoptosis, and overall cellular homeostasis. Furthermore, it integrates other signaling pathways, including TGF-β. The Wnt pathway has two major branches: (i) β-catenin-dependent or canonical and (ii) β-catenin-independent or non-canonical.

AMP: Adenosine monophosphate; HER2: human epidermal growth factor receptor-2; IFN-γ: interferon-gamma; IL-1β: interleukin-1β; IL-6: interleukin-6; TNFα: tumor necrosis factor-α.

**Table 4 biology-12-00963-t004:** *Chlamydia* infection and cancers of different sites.

Investigators, Cancer Site, and Place of Study	Study Design	Results in Brief
Xu et al., 2020 [109] (lung cancer), China	Case-control study, 2006–2016, 449 lung cancer cases and 512 healthy controls.	*C. pneumoniae* specific IgG+ or IgA+ was significantly associated with the increased risk of lung cancer among smokers, persons exposed to passive smoking, and alcohol users.
Chaturvedi et al., 2010 [110] (lung cancer), USA	Case-control study, 593 lung cancers and 671 matched controls.	Individuals with seropositivity for chlamydia heat shock protein-60 (IgG antibodies) had significantly increased lung cancer risk.
Liu et al., 2010 [111] (lung cancer), China	Case–control study, 2000–2009, 192 non-smoking women with lung cancer and 90 healthy controls.	61.98% of cases and 28.89% of controls were positive for *C. pneumoniae* IgG.
Anttila et al., 2003 [112] (lung cancer), Finland	From the Finnish Maternity Cohort, 58 lung cancer cases with pre-diagnostic serum samples, along with 287 control women.	*C. pneumoniae*-specific immune complexes and both IgG and IgA antibodies were associated with female lung cancer.
Bhardwaj et al., 2016 [113] (lymphoma), India	41 ocular adnexal lymphoma cases were analyzed prospectively.	*C. trachomatis* genome was detected in 7.3% cases.
Aigelsreiter et al., 2011 [114] (lymphoma), Austria	Samples from 47 non-G.I. and 14 G.I. MALT lymphomas, 37 nonmalignant controls, and 27 autoimmune precursor lesions were analyzed for the presence of *C. psittaci*, *C. pneumoniae*, and *C. trachomatis* DNA.	13 (28%) non-G.I. MALT lymphomas, 1 (7%) G.I. MALT lymphoma, 4 (11%) nonmalignant control samples, and 11 (41%) autoimmune precursor lesions were positive for *C. psittaci* DNA. No specimens displayed positive results for *C. trachomatis* and *C. pneumoniae*.
Carugi et al., 2010 [115] (lymphoma), Italy	Ocular adnexal lymphoma cases from Italy (n = 30) and Kenya (n = 9).	17% of the samples (from Italy) were positive for *C. psittaci*.
Ferreri et al., 2008 [116] (lymphoma), Italy	20 ocular adnexal MALT lymphoma patients.	*C. psittaci* was detected in lymphoma tissue of 75% patients.
Idahl et al., 2020 [117] (ovarian cancer), Europe	Nested case-control study within the European Prospective Investigation into Cancer and Nutrition (EPIC) cohort, 791 cases and 1669 matched controls.	*C. trachomatis* Pgp3 seropositivity was associated with higher risk of mucinous ovarian carcinoma; chlamydia heat shock protein-60 seropositivity was associated with higher risk of epithelial ovarian cancer (overall) and with the serous subtype.
Fortner et al., 2019 [118] (ovarian cancer), USA	Nested case-control study in the Nurses’ Health Studies (NHS), 337 cases and 337 matched controls.	*C. trachomatis* seropositivity was associated with a two-fold higher risk of ovarian cancer
Trabert et al., 2019 [119] (ovarian cancer), Poland and USA	(i) Polish Ovarian Cancer Study (population-based case–control study conducted in Poland): 244 ovarian cancers and 556 controls. (ii) Prostate, Lung, Colorectal, and Ovarian (PLCO) Cancer Screening Trial (prospective nested case–control study): 160 ovarian cancers and 159 controls.	Antibodies against *C. trachomatis* (Pgp3) were associated with an increased risk of ovarian cancer in these two independent populations.
Ness et al., 2003 [120] (ovarian cancer), USA	Population-based case-control study conducted in Hawaii (1993–1999), 117 ovarian cancer cases and 171 matched controls.	Risk of ovarian cancer was greater in women with higher levels of chlamydia-EB antibodies.
Jensen et al., 2014 [121] (cervical cancer), Denmark	Population-based cohort study; women with high-risk HPV infection and without cervical disease (n = 1390) were followed.	Repeated *C. trachomatis* infections increased the risk of cervical intraepithelial neoplasia grade 3 or worse (CIN3+) in women with prevalent and persistent high-risk HPV infection.
Luostarinen et al., 2013 [122] (cervical cancer), Finland	Cohort consisted of 94,349 women at risk (1995–2003).	Concomitant HPV18/45 and *C. trachomatis* infections were associated with very high risk of cervical intraepithelial neoplasia grade 3 (CIN3).
Olejek et al., 2009 [123] (vulvar cancer), Poland	80 women: 30 vulvar cancer and 50 lichen sclerosus vulvae (risk factor for vulvar cancer).	*C. trachomatis* infection could cause vulvar malignancy.

EB: Elementary body; G.I.: gastrointestinal; HPV: human papillomavirus; Ig: immunoglobulin; MALT: mucosa-associated lymphoid tissue; Pgp3: chlamydia plasmid-encoded protein.

**Table 5 biology-12-00963-t005:** Anti-neoplastic effects of different microbial products and immune checkpoint inhibitors.

Investigators and Studied Toxins	Results (in Brief)
Yang et al. (2013) [144]. Diphtheria toxin-EGF fusion protein (in vitro and in vivo studies)	Fusion protein DAB389EGF treatment inhibited the growth in all tested human bladder cancer cells (J82, RT4, CRL1749, T24, TCCSUP, and HTB9). The intravesical administration of DAB389EGF in female athymic nude mice (nu/nu) with orthotopic xenograft (HTB9) bladder cancer showed a high rate of tumor clearance after 2 weeks of treatment.
Frankel et al. (2002) [145]. USA Diphtheria toxin-GMCSF fusion protein DT388GMCSF (phase I study)	31 patients with acute myeloid leukemia (who were resistant to chemotherapy), 1 had a complete remission and 2 had partial remissions. The main adverse effect was liver injury, and the maximal tolerated dose was 4 μg/kg/day.
Kawai et al. (2021) [146]. Japan Diphtheria toxin fragments A and B and human IL-2 fusion protein E7777 (phase II study)	Patients with relapsed/refractory peripheral T-cell lymphoma (n = 17) and cutaneous T-cell lymphoma (n = 19). Objective response rate was 36%, and the median progression-free survival was 3.1 months. Adverse events: increased AST/ALT, hypoalbuminemia, lymphopenia, and pyrexia.
Ray et al. (2016) [147]. *Vibrio cholera* secreted hemagglutinin protease (in vivo study)	Weekly administration of 1 µg of hemagglutinin protease exhibited potent antitumor activity when injected into Ehrlich ascites carcinoma tumors (murine mammary adenocarcinoma) in Swiss albino mice, and improved the survival rate.
Liu-Chittenden et al. (2015) [148]. USA IL-13-PE: recombinant IL-13 and Pseudomonas exotoxin A (phase I study)	8 patients with adrenocortical carcinoma and distant metastases. Treatment response: 1 patient had stable disease for 5.5 months before disease progression; the others progressed within 1–2 months. Adverse events: low grade anemia, proteinuria, fatigue, and increase in ALT, AST, and creatinine, and development of neutralizing antibodies.
Hassan et al. (2020) [149]. Multicenter-International study LMB-100: anti-mesothelin Fab linked to Pseudomonas exotoxin A (phase I study)	Out of 20 patients, there were no objective partial or complete responses. However, LMB-100 demonstrated an acceptable safety profile.
Hotz et al. (2010) [150]. SLT-VEGF: Shiga-like toxin-VEGF fusion protein (in vivo study)	AsPC-1 and HPAF-2 xenograft pancreatic tumors in nude mice were reduced in the SLT-VEGF group and SLT-VEGF plus gemcitabine combination group.
Monteillier et al. (2018) [151]. Patulin: a mycotoxin from *Penicillium vulpinum* fungus (in vitro study)	In A549 lung cancer cell line, patulin triggered apoptosis, prevented cell migration, and inhibited the Wnt pathway.
Menon et al. (2000) [152]. Cryptophycins: potent anti-microtubule agents isolated from cyanobacteria (in vivo study)	Cryptophycin-52 and cryptophycin-55 combined with doxorubicin, paclitaxel and 5-fluorouracil constitute highly effective regimens against the xenograft MX-1 breast carcinoma in rats. Cryptophycin-52 and cryptophycin-55 were also highly effective against the Calu-6 non-small cell lung carcinoma, H82 and SW-2 small cell lung carcinoma xenografts when combined with the antitumor platinum complexes―cisplatin, carboplatin or oxaliplatin.
Hellmann et al. (2019) [153]. CheckMate-227―global randomized phase III trial. Immune checkpoint inhibitors―nivolumab and ipilimumab	Treatment with nivolumab plus ipilimumab resulted in a longer duration of overall survival in patients with non-small-cell lung cancer compared to the chemotherapy group.
Oh et al. (2022) [154]. South Korea Open-label, single-center, phase II study. Immune checkpoint inhibitors―durvalumab and tremelimumab	124 patients with advanced biliary tract cancer. Gemcitabine and cisplatin plus immune checkpoint inhibitors showed promising efficacy and acceptable safety with regard to adverse events.
Makker et al. (2022) [155]. Multicenter, open-label, randomized, phase III trial. Immune checkpoint inhibitor―pembrolizumab	827 patients with advanced endometrial cancer. Lenvatinib plus pembrolizumab led to significantly longer progression-free survival and overall survival than the chemotherapy group.
Cho et al. (2022) [156]. Global phase II, randomized and blinded study. Immune checkpoint inhibitors―atezolizumab and tiragolumab	135 patients with non-small-cell lung cancer. Tiragolumab plus atezolizumab exhibited a substantial improvement in objective response rate and progression-free survival compared to the placebo plus atezolizumab group.

ALT: alanine aminotransferase, AST: aspartate aminotransferase, EGF: epidermal growth factor, GMCSF: granulocyte-macrophage colony-stimulating factor, IL: interleukin, VEGF: vascular endothelial growth factor. Lenvatinib: kinase inhibitor against the VEGF-receptors; Tiragolumab: binds to T-cell immunoreceptor; Cytotoxic T-lymphocyte-associated antigen (CTLA-4) inhibitors: ipilimumab, and tremelimumab; Programmed cell death/programmed cell death ligand 1 (PD-1/PD-L1) inhibitors: nivolumab, durvalumab, pembrolizumab, and atezolizumab; Objective response rate: the percentage of cases who have a partial response (tumor size decrease) or complete response (disappearance of all cancer signs) to the therapy within a certain period.

## Data Availability

Not applicable.

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
