# Peer review of "An Overview of Selected Bacterial Infections in Cancer, Their Virulence Factors, and Some Aspects of Infection Management"

_biology, 2023, doi:10.3390/biology12070963_

Round 1

Reviewer 1 Report

This review summarized and  associated bacterial infections and neoplasia. This is a relevant subject of study as the association of chronic inflammation and cancer is known, but not yet clarified.

The manuscript could improve with more figures, showing the pathological mechanisms because some sections are quite dense and have many detailed information.

Comments:

(1) Regarding the relationship of H pylori and gastric cancer.

(1a) If a high population of humans are infected by this microorganism symptomatically or assymptomatically, is it possible that the association to the development of neoplasia is just circumstantial?

(1b) Could you please expand the data regarding the induction of NFKB, mutations of TP53, and changes of methylation?

(1b) Regarding citation 21. Could you please confirm the data that 2-3% of infected by h.pylori people will develop gastric adenocarcinoma? This percentage looks too high.

(1c) Could you please expand the description of MALT lymphoma? Clinicopathological, histological, treatment, molecular changes, immune microenvironment, predictors, etc.

(2) Is MALT lymphoma a monoclonal disease? How is it possible that h. pylori erradication reverses neoplasia? H. pylori may trigger chronic inflammation, and this may trigger carcinoma or lymphoma. But, when neoplasia is stablished, should this step be independent of H.pylori? Or is it MALT a "pre-tumoral disease" is some cases? Are there molecular changes associated to resistance to h.pylori erradication?

(3) In the summary it is stated that weakened immune system results in secondary infections. Could you please expand the information regarding immune checkpoint markers and their drug inhibitors? Do changes in the immune regulatory pathways because of cancer also contribute to bacterial infections?

(4) Could you please make a table showing the utilization of bacteria in cancer therapeutics? Are there clinical trials about this topic?

(5) What about human papilloma virus and cancer? What about retrovirus? (the title of the review and topic are bacterial infections, but virus may be of interest as well).

(6) Any data about relationship of vaccinations / infections ---> immune deregulation ---> changes in types of infections ---> chronic inflammation ---> neoplasia?

Author Response

Comments:

(1) Regarding the relationship of H pylori and gastric cancer.

(1a) If a high population of humans are infected by this microorganism symptomatically or assymptomatically, is it possible that the association to the development of neoplasia is just circumstantial?

Our Response: This statement is true for any disease.

All additions/modifications in the revised manuscript are in blue font color.   

(1b) Could you please expand the data regarding the induction of NFKB, mutations of TP53, and changes of methylation?

Our Response: According to the suggestions, the abovementioned phenomena have been elaborated.

(1b) Regarding citation 21. Could you please confirm the data that 2-3% of infected by h.pylori people will develop gastric adenocarcinoma? This percentage looks too high.

Our Response: From the said Reference, the data are correctly cited in our manuscript.  

(1c) Could you please expand the description of MALT lymphoma? Clinicopathological, histological, treatment, molecular changes, immune microenvironment, predictors, etc.

Our Response: According to the suggestions, every aspect has been discussed in the revised manuscript.  

(2) Is MALT lymphoma a monoclonal disease? How is it possible that h. pylori erradication reverses neoplasia? H. pylori may trigger chronic inflammation, and this may trigger carcinoma or lymphoma. But, when neoplasia is stablished, should this step be independent of H.pylori? Or is it MALT a "pre-tumoral disease" is some cases? Are there molecular changes associated to resistance to h.pylori erradication?

Our Response: In light of the current state of knowledge, the aforementioned issues have been discussed. 

(3) In the summary it is stated that weakened immune system results in secondary infections. Could you please expand the information regarding immune checkpoint markers and their drug inhibitors? Do changes in the immune regulatory pathways because of cancer also contribute to bacterial infections?

Our Response: Considering the subject of our review paper, the aforesaid topics have been discussed in Table 5, and in a new section “Systemic cancer therapy – immunity and infection”. We do not want to elaborate on these topics – which may be inappropriate for the present review paper.

(4) Could you please make a table showing the utilization of bacteria in cancer therapeutics? Are there clinical trials about this topic?

Our Response: A Table (Table 5) has been created according to the Reviewer’s suggestions. The findings of a few clinical trials are also mentioned in the Table.  

(5) What about human papilloma virus and cancer? What about retrovirus? (the title of the review and topic are bacterial infections, but virus may be of interest as well).

Our Response: Although the HPV problem has been discussed in Table 4 along with Chlamydia, however, any deviation from bacteria probably is not suitable for the current review paper. Therefore, please allow us to concentrate on the bacterial problems only.             

(6) Any data about relationship of vaccinations / infections ---> immune deregulation ---> changes in types of infections ---> chronic inflammation ---> neoplasia?

Our Response: Some findings have been mentioned in the new section “Systemic cancer therapy – immunity and infection”. There is a limited number of studies (on the subject of the abovementioned suggestions) with regard to the ‘bacterial role and cancer’ topic.   

We are thankful to the Reviewer for the valuable suggestions.  

Reviewer 2 Report

In this manuscript, the authors have mentioned the importance of specific microorganisms and human diseases, e.g gastric cancer. Also, they have discussed the mechanisms the microorganisms could involve in the disease progression. They have also provided information on eradicating microorganisms, which could improve the treatment efficacy.

Comment 1:

However, in this manuscript, the authors did not discuss colon cancer. To enrich the manuscript, the authors should include it as several important findings were identified based on the studies on colon cancer.

Comment 2:

Immunotherapy is becoming more important and common in clinical settings. Would immunotherapy affect the population of the microorganisms and thus the treatment efficacy and side effect? Although solid answers might be absent, the authors might discuss them.

Comment 3:

Table 3 is good. How about making similar tables for other bacterial species and human diseases? 

Author Response

Comment 1:

However, in this manuscript, the authors did not discuss colon cancer. To enrich the manuscript, the authors should include it as several important findings were identified based on the studies on colon cancer.

Our Response: According to the suggestions, a section on colon cancer has been included.

All additions/modifications in the revised manuscript are in blue font color.  

Comment 2:

Immunotherapy is becoming more important and common in clinical settings. Would immunotherapy affect the population of the microorganisms and thus the treatment efficacy and side effect? Although solid answers might be absent, the authors might discuss them.

Our Response: Immunotherapy and relevant bacterial infections will be a vast subject, and many aspects are probably not appropriate for our present review paper. However, according to the suggestions, selected findings have been discussed in a new section “Systemic cancer therapy – immunity and infection”.

Comment 3:

Table 3 is good. How about making similar tables for other bacterial species and human diseases? 

Our Response: Following the suggestions, we have created a similar Table (Table 2) for ‘bacterial infections and colon cancer’.

We are thankful to the Reviewer for the valuable suggestions.  

Round 2

Reviewer 2 Report

The authors have already addressed all my concerns.